# Physical activity, diet and BMI in children aged 6–8 years: a cross-sectional analysis

Laura Basterfield,[1] Angela R Jones,[1] Kathryn N Parkinson,[1] Jessica Reilly,[1] Mark S Pearce,[2] John J Reilly,[3] Ashley J Adamson,[1] The Gateshead Millennium Study Core Team

[1]Institute of Health & Society, Human Nutrition Research Centre, Newcastle University, Newcastle upon Tyne, UK
[2]Institute of Health & Society, Newcastle University, Sir James Spence Institute, Royal Victoria Infirmary, Newcastle upon Tyne, UK
[3]Physical Activity for Health Group, School of Psychological Sciences and Health, University of Strathclyde, Glasgow, UK

**Correspondence to**
Professor Ashley J Adamson;
ashley.adamson@ncl.ac.uk

## ABSTRACT

**Objective:** To assess relationships between current physical activity (PA), dietary intake and body mass index (BMI) in English children.

**Design and setting:** Longitudinal birth cohort study in northeast England, cross-sectional analysis.

**Participants:** 425 children (41% of the original cohort) aged 6–8 years (49% boys).

**Main outcome measures:** PA over 7 days was measured objectively by an accelerometer; three categories of PA were created: 'active' ≥60 min/day moderate-to-vigorous-intensity PA (MVPA); 'moderately active' 30–59 min/day MVPA; 'inactive' <30 min/day MVPA. Dietary intake over 4 days was measured using a prospective dietary assessment tool which incorporated elements of the food diary and food frequency methods. Three diet categories were created: 'healthy', 'unhealthy' and 'mixed', according to the number of portions of different foods consumed. Adherence to the '5-a-day' recommendations for portions of fruit and vegetables was also assessed. Children were classified as 'healthy weight' or 'overweight or obese' (OW/OB) according to International Obesity Taskforce cutpoints for BMI. Associations between weight status and PA/diet categories were analysed using logistic regression.

**Results:** Few children met the UK-recommended guidelines for either MVPA or fruit and vegetable intake, with just 7% meeting the recommended amount of MVPA of 60 min/day, and 3% meeting the 5-a-day fruit and vegetable recommendation. Higher PA was associated with a lower OR for OW/OB in boys only (0.20, 95% CI 0.04 to 0.88). There was no association detected between dietary intake and OW/OB in either sex.

**Conclusions:** Increasing MVPA may help to reduce OW/OB in boys; however, more research is required to examine this relationship in girls. Children are not meeting the UK guidelines for diet and PA, and more needs to be done to improve this situation.

## Strengths and limitations of this study

- Direct objectively measured physical activity and observed dietary intake.
- Direct measures of body composition.
- Population-based birth cohort socioeconomically representative of northeast England.
- Ethnically homogeneous sample which reduces generalisability.
- Cross-sectional so cannot determine direction of causality.

## INTRODUCTION

Childhood obesity (OB) is a major public health challenge in the UK and around the world, with governments increasingly involving themselves in different strategies to reduce excess weight in the population.[1] [2] Recent findings from the National Child Measurement Programme (2011–2012) regarding the prevalence of childhood overweight (OW) and OB are concerning,[3] especially in northeast England, which is the worst-affected region of England outside of London, where 24.5% of 4–5-year-olds and 37% of 10–11-year-olds are OW/OB. There remains a limited understanding of the timing and causes of childhood OB, and there is an urgent need for effective preventive strategies.

Current UK health recommendations state that children should accumulate at least 60 min of moderate-to-vigorous-intensity physical activity (MVPA) per day, with vigorous-intensity activities performed on at least 3 days/week.[4] This recommendation is based on evidence from intervention studies.[5] However, recent evidence suggests that the majority of children fail to meet this target, with 51% of English 4–10-year-old boys and 34% of 4–10-year-old girls achieving the physical activity (PA) recommendations as measured by an accelerometer, figures which dropped to 7% and 0%, respectively, among 11–15-year-olds.[6] In addition, children should consume a healthy balanced diet which

contains a variety of foods, including at least five portions of fruit and vegetables per day, plenty of starchy foods (particularly wholegrain varieties), some protein-rich foods and some milk and dairy foods. It is also recommended that foods high in fat (especially saturated fat), salt and sugar should only be consumed in small amounts.[7] Data from the National Diet and Nutrition Survey in 2010 indicated that children aged 4–10 years consume an average of 173 g of fruit and vegetables daily, equivalent to 2.2 80 g portions (or 2.8 including fruit juice).[8] Data from longitudinal cohorts are beginning to uncover the associations between low levels of PA and unfavourable body composition, for example, Janz et al,[9] but data on the relationship between diet/dietary components and body composition have so far been equivocal, as reviewed by Must et al.[10] This may be due to the contrasting methods of dietary assessment and analysis and the complexity of dietary patterns; given the importance of energy balance in maintenance of a healthy body weight, it is vital that measurement of dietary intake continues to be improved. Evidence of the impact of objectively measured PA/sedentary behaviour (SB) and food intake on body composition in primary school-age children is lacking, with most studies measuring either PA or food intake. A recent Scottish study assessed the effect of dietary patterns and PA on body mass index (BMI),[11] but used a questionnaire that is known to overestimate time spent in PA.[12] The authors did not find a consistent association between dietary patterns or time spent in PA and BMI group.[11]

The primary aim of the present study was to describe the relationship between OW and OB, BMI, PA and dietary habits in a sample of children that are socio-economically representative of northeast England.[13]

## METHODS
### Participants
The Gateshead Millennium Study (GMS) cohort originally consisted of 1029 babies and their families and is described in detail elsewhere.[13] Briefly, all mothers resident in Gateshead who gave birth within prespecified recruiting weeks between June 1999 and May 2000 were invited to take part; 81% (n=1011) agreed. Participants were mostly from the white ethnic majority (98%). Data collection for the present study involved a variety of lifestyle and anthropometric measures, and took place between October 2006 and December 2007 when the children were aged 6–8 years. The data presented are cross-sectional from the 6–8-year collection phase of the GMS. Parents provided informed written consent for their child's participation and children gave their written assent.

### Measures
#### PA and SB
PA and SB were objectively measured using Actigraph GT1M accelerometers (Actigraph LLC, Pensacola, Florida, USA) attached to an elastic belt and worn on the right hip as described previously.[14] Children were asked to wear the accelerometer during waking hours for 7 days and to remove it only for water-based activities. Parents were asked to complete a timelog detailing when the accelerometer had been worn.

The accelerometers collected data in 15 s sampling intervals (epochs); prior to analysis, the data were collapsed to 60 s epochs. Data were verified manually using the timelog and accelerometry output as described previously[12] to delete occasional periods of non-wear time.

Three constructs of PA and one of SB were generated: total volume of activity (expressed as mean daily accelerometer counts per minute, cpm), MVPA daily, equivalent to energy expenditure in excess of approximately three times resting energy expenditure (expressed as minutes/day and percentage of daily time spent in MVPA) and SB (equivalent to no translocation of the trunk[15] (expressed as percentage of daily time spent in SB)).

The cutpoint of $\geq 3200$ cpm[16] was used to define MVPA and $<1100$ cpm[17] was used to define SB. The Actigraph GT1M model has been shown to have a consistent 9% bias relative to model 7164[18] from which the cutpoints were derived, so a +9% correction was applied to the raw accelerometry data before applying cutpoints. Of 606 accelerometers given out, 510 were worn for at least 3 days and for at least 6 h/day[19] and thus were deemed eligible for analysis in the present study.

### Dietary intake
Food intake in the children was assessed over 4 days (2 weekdays and 2 weekend days) using the 'Food Assessment in Schools Tool' (FAST) food diary.[20] The FAST food diary is a prospective dietary assessment method which incorporates elements of the food diary and the food frequency methods and was specifically designed to assess the diets of primary school children. Briefly, each day is split into six timeslots, each containing a simple tick list of up to 50 foods commonly eaten by children of this age.[21] Space for additional foods was also provided. Researchers and lay observers recorded food intake for each child during the school day, and parents and other carers recorded food intake at all other times. No interviews or estimations of portion size were required during data collection; average age-specific and sex-specific portion sizes for all foods consumed were ascertained using relevant UK National Diet and Nutrition Survey data.[20] [21] This method allowed analysis of food choice at an individual level but not of energy intake.

A 'healthy' and 'unhealthy' dietary indicator score was created for each of the 4 days, and a daily mean calculated. The 'healthy' indicator (HI) score was calculated by summing the number of portions consumed of the following foods: wholemeal bread; non-high sugar wholegrain breakfast cereals; fruit; vegetables; semi-skimmed milk as a drink; semi-skimmed milk; non-processed meat and fish products; beans and pulses; water and white bread with added fibre. The 'unhealthy'

indicator (UI) score was calculated by summing the number of portions consumed of the following foods: white bread; high-sugar refined breakfast cereals; fried chips; biscuits; confectionery, cakes and sweet puddings; full-fat milk as a drink; full-fat milk; processed meat and fish products; full-sugar cordial or squash; full-sugar carbonated drinks; chocolate/milkshake powder; crisps and savoury snacks and sugar. A cereal was categorised as 'non-high sugar' if it contained <5 g sugar/100 g and 'high sugar' if it contained >15 g sugar/100 g as manufactured. The addition of discretionary sugar was not taken into account when categorising cereals. To assess adherence to dietary guidelines, fruit and vegetable consumption was measured in 80 g amounts, fruit juice in 150 mL portions. Only one portion of fruit juice may count towards daily fruit and vegetable intake.[7]

### Anthropometry
All body measurements were taken in duplicate with the children in light indoor clothing, without socks and shoes. Height was measured to 0.1 cm with a Leicester portable height measure (Chasmors, London, UK), waist to 0.1 cm at the minimum circumference between the lowest rib and iliac crest, and weight to 0.1 kg using a TANITA TBF 300MA body fat analyser (Chasmors, London, UK). BMI was calculated as (wt (kg)/ht (m$^2$)). Categories of underweight, healthy weight, OW and OB were calculated according to age-specific and sex-specific International Obesity Taskforce (IOTF) criteria.[22 23]

### Statistical analysis
The sample size for the current cross-sectional study was fixed by the size of the cohort, participant attrition and those children providing complete data for PA, dietary intake and anthropometry at this data sweep.

Analysis was performed using SPSS V.17 and STATA V.10. Shapiro-Wilk tests assessed the normality of distribution of variables. As data were skewed, Mann-Whitney U tests were performed to test for differences between sexes and Wilcoxon rank tests to assess differences in weekend and weekday PA/SB. Median and IQRs were reported unless otherwise stated.

Logistic regression was used to predict the likelihood of being OW/OB at different levels of PA and with different dietary indicator scores. Children were assigned to one of the three PA categories: (1) 'active': those who met the ≥60 min/day guideline (this figure was generated by dividing the total amount of MVPA for the week by the number of days the accelerometer was worn); (2) 'moderately active': between 30 and 60 min/day; (3) 'inactive': less than 30 min/day. Children were also assigned to one of the three dietary categories: (1) 'healthy': those children whose HI score was ≥the median and whose UI score was < the median; (2) 'unhealthy': those children whose UI score was ≥the median and whose HI score < the median; (3) 'mixed': those diets which neither met the criteria for a 'healthy' diet nor for an 'unhealthy' diet. Children were lastly classified by BMI as: (1) not OW/OB; (2) OW/OB. ORs and corresponding 95% CIs were reported. Significance was set at p<0.05. Sex was included in the regression analysis as a potential confounder. Townsend score from birth (a measure of deprivation) was unrelated to any of the variables and thus not included in the final analysis.

### RESULTS
Overall, 510 children completed sufficient accelerometry recording; 474 children returned completed FAST food diaries and 597 children had anthropometry measurements. Data from all three domains were provided by 425 children and were used in the analysis. There were no significant differences in food intake, PA or BMI between children providing complete records and those who did not. Children had a mean age of 7.4 years (range 6.4–8.4). Characteristics of the sample are displayed in table 1.

Using IOTF cut-offs,[22] 24% of the children were OW/OB, with no significant difference between girls and boys (Mann-Whitney U test, p=0.885). The full distribution of children between weight categories is shown in table 2.

There were no significant differences between the sexes in the number of days the activity belt was worn or the total volume of activity (mean cpm; table 3).

**Table 1**  Participant characteristics (median and IQR), by sex

| | Combined*<br>(n=425) | | Boys (n=211)<br>(49.6%) | | Girls (n=214)<br>(50.4%) | | p Value |
|---|---|---|---|---|---|---|---|
| Height (cm) | 1.24 | 1.21, 1.29 | 1.26 | 1.22, 1.29 | 1.24 | 1.21, 1.28 | *0.039* |
| Weight (kg) | 26.4 | 22.7, 28.7 | 25.7 | 23.0, 29.1 | 25.0 | 22.5, 28.4 | 0.264 |
| Waist (cm) | 55.7 | 52.6, 59.0 | 56.5 | 53.5, 59.4 | 54.8 | 52.1, 58.7 | *0.006* |
| BMI (kg/m$^2$) | 16.2 | 15.3, 17.8 | 16.1 | 15.4, 17.8 | 16.3 | 15.2, 17.8 | 0.878 |

*n For individual measures varying from 421 to 425. Mann-Whitney U tests performed.
BMI, body mass index.
Italic font indicates significant results.

**Table 2** Distribution of children between weight categories, by sex

|  | Combined | | Boys (49.6%) | | Girls (50.4%) | |
|---|---|---|---|---|---|---|
|  | n | Per cent | n | Per cent | n | Per cent |
| Underweight | 3 | 0.7 | 1 | 0.5 | 2 | 0.9 |
| Healthy weight | 320 | 75.3 | 160 | 75.8 | 160 | 74.8 |
| Overweight | 75 | 17.6 | 35 | 16.6 | 40 | 18.7 |
| Obese | 27 | 6.4 | 15 | 7.1 | 12 | 5.6 |
| Total | 425 | 100.0 | 211 | 100.0 | 214 | 100.0 |

Levels of activity in general were low, as reported previously;[12] only 31 children (7%) achieved an average of at least 60 min MVPA/ day, 23 boys and 8 girls. Only two children (0.5%) accumulated at least 60 min on each day they wore the accelerometer. The majority of time was spent in SB (median 78% of wear time, equivalent to around 8.6 h/day). When looking at the differences between weekend and weekday activity, volume of activity (cpm) was similar, and slightly but significantly more MVPA was taken on weekdays (p<0.0001; data not shown). Girls had a greater volume of activity on weekends but a lower time spent in MVPA, suggesting that the majority of their activities are in the sedentary or light intensity categories. Boys spent significantly more time in MVPA on weekdays than girls.

Dietary intake was similar for boys and girls (table 3). Children consumed a median of 1.7 portions of fruit and vegetables daily (or 2.1 portions when including fruit juice), with just 13/425 children (3.1%) consuming '5 a day'. On average, children ate 5.0 HI foods and 5.3 UI foods daily.

Table 4 displays the distribution of OW/OB by diet and activity group.

13% of healthy weight boys were in the 'active' PA category, compared with just 4% of OW/OB boys. 48% of healthy weight boys were inactive compared with 76% of OW/OB boys. There was less variation for girls, as only eight girls in total were in the active category. In total, 59% of healthy weight girls were in the inactive category compared with 71% of OW/OB girls.

Compared with the reference group (inactive), children in the moderately active group had a significantly lower risk of being OW/OB (OR=0.44, 0.26 to 0.74 (table 5)).

**Table 3** Comparison of physical activity, sedentary behaviour (SB) and dietary intake between sexes (median and IQR unless otherwise specified)

|  | Combined (n=425) | | Boys (n=211) | | Girls (n=214) | | p Value |
|---|---|---|---|---|---|---|---|
| **Physical activity/SB** | | | | | | | |
| Number of days accelerometer worn | 7 | 6, 7 | 7 | 6, 7 | 7 | 6, 7 | 0.380 |
| Hours worn/day | 11.3 | 10.6, 11.9 | 11.4 | 10.7, 12.0 | 11.1 | 10.4, 11.8 | 0.077 |
| Mean cpm/day | 727 | 604, 878 | 740 | 624, 881 | 718 | 591, 868 | 0.313 |
| MVPA (min/day) | 26 | 18, 38 | 28 | 19, 43 | 25 | 16, 36 | *0.020* |
| Percentage of time in MVPA | 3.9 | 2.6, 5.7 | 4.2 | 2.7, 6.3 | 3.8 | 2.5, 5.5 | *0.043* |
| Percentage of time SB | 77.7 | 73.6, 81.8 | 76.9 | 71.9, 80.8 | 78.4 | 74.6, 82.7 | *0.002* |
| Achieved 60 min/day MVPA (n, %)* | 31 | 7.3 | 23 | 10.9 | 8 | 3.7 | *0.005* |
| **Dietary intake** | | | | | | | |
| Portions of fruit and vegetables consumed/day | 1.7 | 1.1, 2.5 | 1.7 | 1.0, 2.5 | 1.8 | 1.2, 2.5 | 0.211 |
| Portions of fruit and vegetables consumed/day† | 2.1 | 1.4, 3.0 | 2.0 | 1.3, 2.9 | 2.2 | 1.4, 3.1 | 0.224 |
| Ate 5 portions of fruit and vegetables/day (n, %)† | 13 | 3.1 | 8 | 3.8 | 5 | 2.3 | 0.384 |
| Number of healthy indicator foods/day | 5.0 | 3.8, 6.5 | 5.0 | 3.8, 6.5 | 5.1 | 3.8, 6.5 | 0.342 |
| Number of unhealthy indicator foods/day | 5.3 | 4.3, 6.3 | 5.5 | 4.3, 6.3 | 5.3 | 4.3, 6.3 | 0.496 |

*Total MVPA for the week/days worn.
†Including one 150 mL portion of juice. Mann–Whitney U tests performed.
SB, sedentary behaviour; MVPA, moderate-to-vigorous-intensity physical activity.
Italic font indicates significant results.

**Table 4** Distribution of OW/OB by PA and dietary category (n, %)

| PA category | Healthy weight (323, 76.0) Dietary pattern | | | | Overweight or obese (102, 24.0) Dietary pattern | | | |
| | Unhealthy | Mixed | Healthy | Total | Unhealthy | Mixed | Healthy | Total |
| --- | --- | --- | --- | --- | --- | --- | --- | --- |
| Boys (211, 49.6) | | | | | | | | |
| Inactive | 27, 16.8 | 33, 20.5 | 18, 11.2 | 78, 48.4 | 11, 22.0 | 13, 26.0 | 14, 28.0 | 38, 76.0 |
| Moderately active | 17, 10.6 | 28, 17.4 | 17, 10.6 | 62, 38.5 | 2, 4.0 | 7, 14.0 | 1, 2.0 | 10, 20.0 |
| Active | 9, 5.6 | 10, 6.2 | 2, 1.2 | 21, 13.0 | 0, 0.0 | 1, 2.0 | 1, 2.0 | 2, 4.0 |
| Total | 53, 32.9 | 71, 44.1 | 37, 23.0 | 161, 100.0 | 13, 26.0 | 21, 42.0 | 16, 32.0 | 50, 100.0 |
| Girls (214, 50.4) | | | | | | | | |
| Inactive | 27, 16.7 | 29, 17.9 | 40, 24.7 | 96, 59.3 | 9, 17.3 | 18, 34.6 | 10, 19.2 | 37, 71.2 |
| Moderately active | 17, 10.5 | 31, 19.1 | 12, 7.4 | 60, 37.0 | 4, 7.7 | 4, 7.7 | 5, 9.6 | 13, 25.0 |
| Active | 1, 0.6 | 3, 1.9 | 2, 1.2 | 6, 3.7 | 0, 0.0 | 2, 3.8 | 0, 0.0 | 2, 3.8 |
| Total | 45, 27.8 | 63, 38.9 | 54, 33.3 | 162, 100.0 | 13, 25.0 | 24, 46.2 | 15, 28.8 | 52, 100.0 |

OW/OB, overweight or obese; PA, physical activity.

Results for children in the active group (≥60 min MVPA/ day, OR=0.34, 95% CI 0.12 to 1.02) approached significance (p=0.054). Similar results were seen after adjusting for sex. When analysing girls and boys separately, the association between PA and risk of OW/OB remained only for boys. Boys who were moderately active had an OR of 0.33 (0.15 to 0.72) for risk of OW/OB (p=0.005), and boys in the active category had an OR of 0.20 (0.04 to 0.88; p=0.033), compared with the inactive group. Moderately active girls had an OR of 0.56 (0.28 to 1.14; p=0.112) and active girls an OR of 0.86 (0.17 to 4.48; p=0.863) for OW/OB compared with inactive girls. Similar results were seen for activity when adjusting for diet group.

There was no influence of dietary pattern on risk of OW/OB: OR 1.27 (0.73 to 2.19; p=0.40) for the mixed diet group, 1.28 (0.71 to 2.33) for the healthy group (p=0.41) compared with the reference group (unhealthy diet; table 5), with similar results after adjusting for sex.

There was no evidence of an interaction between PA and food intake on risk of OW/OB (p=0.444).

## DISCUSSION

Our results provide evidence that children in northeast England are not meeting the guidelines for PA and nutrition that are recommended for good health. Lack of MVPA was associated with an increased risk of being OW and OB in boys. The lack of association in girls

could be due to the low numbers being sufficiently active.

The present study has limitations in terms of the ethnically homogeneous sample, which may limit the generalisability of the findings to other samples and settings. There was also a lack of variability in PA, with data skewed towards inactivity, particularly in girls. There are several cutpoints in use for classifying MVPA[24] and the value of 3200 cpm used in the current study was obtained through comparison with energy expenditure,[16] and similar to those from other studies.[15] Given the ongoing discussion regarding the choice of cutpoint, there may be no suitable cutpoint for all ages of childhood. As the data are cross-sectional, it is not possible to determine the direction of causality, which will be addressed by future follow-up. However, the strengths of this study include the relatively large sample size, the focused age of the children, and the use of objective measures of PA (accelerometry) combined with dietary observations is unique.

Few other studies have used objective methods to describe levels of habitual PA and SB in 6–8-year-olds in combination with dietary intake measures. Recent UK studies in 3–5-year-olds[14] and 11–12-year-olds[25] which have used the same objective methods of PA measurement have found similarly low levels of PA with adherence to the 60 min/day recommendation of MVPA being achieved in <5% of their samples. In contrast, UK studies which have used subjective methods (questionnaires) have generally

**Table 5** Likelihood of being overweight or obese for different activity levels and dietary patterns (ORs and 95% CI)

| | Combined (n=425) | p Value | Boys (n=211) | p Value | Girls (n=214) | p Value |
| --- | --- | --- | --- | --- | --- | --- |
| Inactive (reference) | 1 | | 1 | | 1 | |
| Moderately active | 0.44 (0.26 to 0.74) | *0.002* | 0.33 (0.15 to 0.72) | *0.005* | 0.56 (0.28 to 1.14) | 0.112 |
| Active | 0.34 (0.12 to 1.02) | 0.054 | 0.20 (0.04 to 0.88) | *0.033* | 0.86 (0.17 to 4.48) | 0.863 |
| Unhealthy diet (reference) | 1 | | 1 | | 1 | |
| Mixed diet | 1.27 (0.73 to 2.19) | 0.40 | 1.21 (0.55 to 2.62) | 0.637 | 1.32 (0.61 to 2.86) | 0.485 |
| Healthy diet | 1.28 (0.71 to 2.33) | 0.41 | 1.76 (0.76 to 4.10) | 0.188 | 0.96 (0.42 to 2.23) | 0.927 |

Italic font indicates significant results.

reported much higher levels of MVPA, for example, the Health Survey for England, which leads to overestimates of PA.[12] Low levels of PA were present in our sample well before puberty, the period traditionally regarded as being associated with low levels of PA. In keeping with other studies,[9 26] we found that boys were more active than girls, and the lack of association of PA on risk of OW/OB in girls as compared with boys suggests further work is required into the mechanisms by which increasing PA and/or decreasing SB reduces risk of OW/OB in children. Few studies have looked at the relationship between the '60 min' guidelines and body composition. Martinez-Gomez et al[27] reported that 60 min/day MVPA could discriminate between normal weight and OW/OB adolescents (mean age 14.7 years), however they used a lower cutpoint for MVPA of 2000 cpm, compared with the 3200 cpm used in the present study. This resulted in average MVPA/ day of 66 min for boys and 50 min for girls.[27] Recent evidence using 3200 cpm as the MVPA cutpoint for 10-year-olds in France[28] found that only 10% of the 'normal weight' children reached 60 min/day, but 44% did at least 30 min/day. In addition, although only 5% of OW/OB children did at least 60 min/day MVPA, 34% reached 30 min/day.[28] Our results show that this lower recommendation could still have benefits in terms of body composition, particularly for boys.

While it is generally accepted that PA decreases with age,[29] pinpointing the age when the decrease starts is important for targeted interventions. Our results suggest that even at the age of 7 years, PA levels are far lower than recommended. The great proportion of time spent sedentary by both sexes is concerning—in both sexes almost 7 h/day were spent with no translocation of the trunk.[15] Both sexes took part in more MVPA on weekdays than weekends, highlighting the role the school environment plays in maintaining activity levels, and that more needs to be done to increase activity on weekends. The cross-sectional nature of the present study means we cannot say whether the children have already started to reduce their activity. However, data we have published from the cohort at a 2-year follow-up show a decrease in MVPA from age 7 to 9 years, and the decrease in MVPA associated with an increase in BMI-z score in boys.[30 31]

The low consumption of fruit and vegetables, and the relatively high consumption of an unhealthy diet in this sample is cause for concern, despite the lack of association between the 'healthy' or 'unhealthy' categories and risk of OW/OB. The lack of association between dietary variables and risk of OW/OB in this study, as in others, may be due to the lack of individual portion sizes which prohibited the assessment of energy intake. The dietary assessment tool used in this study allowed direct observation of intake but applied standard portion sizes from a previous National Diet and Nutrition Survey which used a weighed intake method.[21] The amount for a 'portion' of fruit and vegetables was set at 80 g, which is considered suitable for children aged over 11 years.[32] A lower amount may be more suitable for children under 11 years, but as this has not yet

been identified,[32] 80 g was used in this analysis. It is possible that OW children are eating larger portions of healthy foods, or the parents are aware of their child's OW and have reduced food intake accordingly. There may also be an element of misreporting or optimistic bias if the parent believes their child is OW and eats either to excess or 'unhealthily'. Foods in the current study were assigned to the 'healthy' or 'unhealthy' group based on a combination of current national guidelines for consumption and restriction[7] according to nutritional content. Since these analyses were performed, Frémeaux et al[33] have published a longitudinal factor analysis of English children's food intake from age 5 to 13 years. They reported the emergence of two main diet types that matched almost identically with ours.[33] Recent evidence from Scottish 5–11-year-olds did not find an association between dietary patterns and BMI or PA,[11] and observed an inverse association between 'screen time' and healthier eating patterns (diet measured with food frequency questionnaire (FFQ) and PA measured by Physical activity questionnaire, PAQ). The Avon Longitudinal Study of Parents and Children (ALSPAC) cohort in the UK was assessed for associations between PA and diet at age 10–11 years, and found only weak relationships between the two[34] (3×1-day food diary at 10 years, accelerometry at 11 years). A Greek case–control study found a protective effect of PA[35] on risk of OW/OB, and an increased risk with increased time spent watching TV, and consumption of sugar-sweetened beverages (FFQ/PAQ), an observation also reported in Spanish children (FFQ/PAQ).[36] Others[37] found an effect of SB on risk of OW/OB, but again no effect of diet composition (questionnaire). A varied food pattern plus a physically active lifestyle was inversely related to OW in French children (7-day food diary/PAQ).[38] Similarly, reduced PA, but not excessive energy intake, was associated with increased risk of OB in Turkish children (3-day food diary/PAQ).[39] However, these studies included a wide age range of children, and PA and SB were measured objectively only in the ALSPAC cohort,[34] so our observations strengthen the evidence base for the protective effects of PA on risk of OW/OB.

In conclusion, children in northeast England have extremely low levels of habitual MVPA, high levels of SB and consume a diet low in fruit and vegetables but high in foods associated with an unhealthy diet. Increased PA was associated with reduced risk of OW/OB in boys. Further work is required to uncover the associations of PA/diet and OW/OB in girls and the longitudinal associations between PA/diet and OW/OB in boys and girls. These observations give cause for concern in relation to their current and future risk of cardiovascular diseases, diabetes and OB. Our observations suggest that UK government targets in relation to levels of PA and nutrition among children are not being met.

**Acknowledgements** The authors acknowledge the support of an External Reference Group in conducting the study. They appreciate the support of Gateshead Health NHS Foundation Trust, Gateshead Education Authority and local schools. They also thank the research team for their effort. Thanks are especially due to the Gateshead Millennium Study families and children for their participation in the study.

**Collaborators** The Gateshead Millennium Study Core Team: AJA, Anne Dale, Robert Drewett, Ann Le Couteur, Paul McArdle, KNP, MSP, JJR and Charlotte Wright.

**Contributors** AJA and KNP were responsible for the study conception and design, manuscript drafting and revision. AJA is the study guarantor. JJR and MSP were responsible for the study conception and design, analysis and interpretation of the data, manuscript drafting and revision. LB, ARJ and JR were responsible for data acquisition, analysis and interpretation of the data; manuscript drafting and revision. All authors have had full access to all of the data in the study and take responsibility for the integrity of the data and the accuracy of the data analysis.

**Funding** This work was supported by the National Prevention Research Initiative (NPRI) grant number GO 501306. The Gateshead Millennium Study is supported by a grant from the NPRI (incorporating funding from British Heart Foundation; Cancer Research UK; Department of Health; Diabetes UK; Economic and Social Research Council; Food Standards Agency; Medical Research Council; Research and Development Office for the Northern Ireland Health and Social Services; Chief Scientist Office, Scottish Government Health Directorates; Welsh Assembly Government and World Cancer Research Fund). The cohort was first established with funding from the Henry Smith Charity and Sport Aiding Research in Kids (SPARKS) and followed up with grants from Gateshead NHS Trust R&D, Northern and Yorkshire NHS R&D, and Northumberland, Tyne and Wear NHS Trust.

**Competing interests** None.

**Ethics approval** Gateshead and South Tyneside Local National Health Service Research Ethics Committee.

**Provenance and peer review** Not commissioned; externally peer reviewed.

**Data sharing statement** No additional data are available.

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
