## [Reviewer comments · BMJ Open]

Some articles will have been accepted based in part or entirely on reviews undertaken for other BMJ Group journals. These will be reproduced where possible.

ARTICLE DETAILS

TITLE (PROVISIONAL)	Physical activity, diet and BMI in 6-8 year old children; a cross-sectional analysis
AUTHORS	Basterfield, Laura; Jones, Angela; Parkinson, Kathryn; Reilly, Jessica; Pearce, Mark; Reilly, John; Adamson, Ashley

VERSION 1 - REVIEW

REVIEWER	Giacomo Lazzeri Department of Molecular and Developmental Medicine- University of Siena
REVIEW RETURNED	18-Mar-2014

GENERAL COMMENTS	The statistical analysis, even if no errors, requires a revision in some of its parts, in particular: 1) In the section "statistical analysis", there is no reference to the tests used to verify the normality of distribution of variables2) Before making the non-parametric tests, was probably tempted some data transformation?3) In the notes to Table 2 it is said "with no significant difference between boys and girls" without making any reference to the test used. Council to include the results, including the p-values in Table 2 and the type of test in the section on statistical analysis4) In table 4 would insert also the absolute frequencies, in addition to the percentages5) In Table 5 would recommend to also report the results for the confounding factor and factor interaction
--

REVIEWER	Carrie Ruxton PhD RD Nutrition Communications UK
REVIEW RETURNED	27-Mar-2014

GENERAL COMMENTS	pg 5. Sentence could be reworded slightly to reduce the number of mentions of 'data': "The data presented here are cross-sectional data from this data collection phase only." Methods pg 5: there is no need to abbreviate the Gateshead Millennium Study as you don't use the abbreviation elsewhere in the paper.
---

	Methods pg 6-7: Define high-sugar and non-high sugar refined breakfast cereals. Would sugar added to porridge count? Methods Anthropometry: Were the BMI results compared to age-sex specific standards? If so, please state when you mention the IOTF criteria. Table 3 is a bit confusing as you have different measurements. For example, should "Ate 5 portions of fruit & vegetables/d" be percentage of children? Please check this table and ensure full clarity of what the units are for each measure. Also, could you put the significant P values in italics so they are quicker to assess at a glance? Otherwise, I am pleased with the quality and substance of this paper.
--	--

VERSION 1 – AUTHOR RESPONSE

Reviewer 1

The statistical analysis, even if no errors, requires a revision in some of its parts, in particular:

1) In the section "statistical analysis", there is no reference to the tests used to verify the normality of distribution of variables.

We have amended the statistical analysis section of the manuscript to describe the tests used to assess the normality of distribution of variables (page 7).

2) Before making the non-parametric tests, was probably tempted some data transformation?

We did not transform the data prior to analysis, we took statistical advice on the analysis process and we were informed that performing non-parametric tests would remove the need to back transform the data and would make the results more translatable.

3) In the notes to Table 2 it is said "with no significant difference between boys and girls" without making any referment to the test used. Council to include the results, including the p-values in Table 2 and the type of test in the section on statistical analysis.

We thank the reviewer for highlighting this issue; we have added further detail about the test used and the p value on page 10. We have also clarified the data in Table 2 which is descriptive data of the distribution of children between the weight categories.

4) In table 4 would insert also the absolute frequencies, in addition to the percentages

We have added the frequencies to Table 4 as requested (page 13), percentages are now given in italics.

5) In Table 5 would recommend to also report the results for the confounding factor and factor interaction

There was no evidence of confounding and so we have added in a sentence on page 16 ('Similar results were seen for activity when adjusting for diet group') to add clarification. We have also added the p value when describing the lack of evidence of an interaction between physical activity and food intake on risk of overweight/obesity (page 16).

Reviewer 2

pg 5. Sentence could be reworded slightly to reduce the number of mentions of 'data': "The data presented here are cross-sectional data from this data collection phase only."

Methods pg 5: there is no need to abbreviate the Gateshead Millennium Study as you don't use the abbreviation elsewhere in the paper.

We have amended the sentence on page 5 to remove the number of mentions of 'data' as requested, within this amendment we have used the abbreviation 'GMS' and so have retained the abbreviation on first line of the 'Participants' section.

Methods pg 6-7: Define high-sugar and non-high sugar refined breakfast cereals. Would sugar added to porridge count?

On page 7, paragraph 1, we have added additional detail regarding the cut offs used to define 'high' and 'non-high' sugar cereals. Sugar added to porridge would not count and so we have clarified this by stating "The addition of discretionary sugar was not taken into account when categorising cereals".

Methods Anthropometry: Were the BMI results compared to age-sex specific standards? If so, please state when you mention the IOTF criteria.

BMI results were compared to age- and sex-specific standards and we have added further information on page 7, paragraph 2, when describing the IOTF criteria as requested.

Table 3 is a bit confusing as you have different measurements. For example, should "Ate 5 portions of fruit & vegetables/d" be percentage of children? Please check this table and ensure full clarity of what the units are for each measure.

We thank the reviewer for highlighting this issue, all variables are presented as median and IQR except for two variables ('Achieved 60 min/d MVPA' and 'Ate 5 portions of fruit & vegetables/d'), we have added further detail to the title of the table and within the table and have formatted the table to improve the clarity of the units for each measure.

Also, could you put the significant P values in italics so they are quicker to assess at a glance? Significant p values are now in italics in all tables and throughout the text.